# Heat Stress Induces Alterations in Gene Expression of Actin Cytoskeleton and Filament of Cellular Components Causing Gut Disruption in Growing–Finishing Pigs

**DOI:** 10.3390/ani14172476

**Published:** 2024-08-26

**Authors:** Yohan Choi, Hyunju Park, Joeun Kim, Hyunseo Lee, Minju Kim

**Affiliations:** 1Swine Science Division, National Institute of Animal Science, Rural Development Administration, Cheonan 31000, Republic of Korea; cyh6150@korea.kr (Y.C.); guswn707@korea.kr (H.P.); kjektw@korea.kr (J.K.); 2School of Animal Life Convergence Science, Hankyong National University, Anseong 17579, Republic of Korea; gustjekek@naver.com; 3Institute of Applied Humanimal Science, Hankyong National University, Anseong 17579, Republic of Korea

**Keywords:** heat stress, pigs, intestinal permeability, gut integrity, tight junction, actin cytoskeleton

## Abstract

**Simple Summary:**

Acute heat stress in animals impairs the intestinal barrier function, particularly tight junction (TJ) depression, which occurs in pigs owing to the lack of functional sweat glands for evaporative heat loss. However, the mechanisms by which heat stress affects TJ regulation are not fully understood. We investigated the effect of heat stress on the TJ and its interaction with potential genes using transcriptomic analysis of the porcine jejunum. We found that heat stress reduced the expression of TJ proteins and induced alterations in genes involved in the actin cytoskeleton and filament-binding pathways.

**Abstract:**

We aimed to investigate the impact of heat stress (HS) on the expression of tight junction (TJ) proteins and the interaction between genes affecting intestinal barrier function using transcriptomics in the porcine jejunum. Twenty-four barrows (crossbred Yorkshire × Landrace × Duroc; average initial body weight, 56.71 ± 1.74 kg) were placed in different temperatures (normal temperature [NT]; HS) and reared for 56 days. At the end of the experiment, jejunal samples were collected from three pigs per treatment for transcriptome and reverse-transcription quantitative polymerase chain reaction (RT-qPCR) analyses. We identified 43 differentially expressed genes, involving five Kyoto Encyclopedia of Genes and Genomes pathways, eight molecular functions, seven cellular components (CCs), and nine biological processes, using gene ontology enrichment analysis. Genes associated with the actin cytoskeleton, filament-binding pathways, and TJ proteins were selected and analyzed by RT-qPCR. Significant differences in relative mRNA expression showed that downregulated genes in the HS group included *ZO1*, *CLDN1*, *OCLN*, *PCK1*, and *PCK2*, whereas *ACTG2*, *DES*, *MYL9*, *MYLK*, *TPM1*, *TPM2*, *CNN1*, *PDLIM3*, and *PCP4* were upregulated by HS (*p* < 0.05). These findings indicate that HS in growing-finishing pigs induces depression in gut integrity, which may be related to genes involved in the actin cytoskeleton and filaments of CC.

## 1. Introduction

As global warming accelerates worldwide, the adverse effects of heat stress (HS) on the livestock industry has received considerable attention [1,2]. HS manifests when the ambient temperature exceeds the thermoneutral zone of an animal, and the physiological mechanisms for thermoregulation, such as perspiration, respiration, or panting, are insufficient to prevent elevation in body temperature [3,4]. Pigs, in particular, are sensitive to HS as they lack functional sweat glands for evaporative heat loss [5]. The consequences of HS on swine production are reduced feed intake and growth rate and increased mortality, resulting from intricate changes in morphological and physiological functioning of the gastrointestinal tract (GIT) [6,7].

Higher temperatures induce the dispersion of blood to the periphery for heat dissipation, which results in reduced blood supply and decreased nutrient flow in the intestinal epithelium [8,9]. Consequently, HS impairs intestinal integrity and function, thereby increasing intestinal permeability and susceptibility to infectious diseases [10]. Intestinal permeability, regulated by tight junction (TJ) proteins such as zonula occludens-1 (*ZO1*), occludin (*OCLN*), and claudins (*CLDNs*), is responsible for translocation of antigens, toxins, and pathogens. This stimulates the immune system, and affects nutrient digestibility and absorption across the intestinal epithelium [11,12].

Transmembrane proteins (TJ proteins) that constitute the structure of a TJ interact with the actin cytoskeleton within the cell to regulate the TJ [13]. Together with adherens junctions, TJ proteins are closely associated with the perijunctional actomyosin ring, a belt-like structure composed of actin and myosin II that encircles the apical pole of epithelial cells. This belt extends the actin filaments that interact with the TJ, thereby allowing circumferential contractions of the perijunctional actomyosin ring, thereby regulating TJ architecture and paracellular permeability [14]. However, the mechanisms underlying the alteration of intestinal permeability and the interaction with TJ regulation in the actin cytoskeleton and filament pathways of pigs exposed to HS are not fully understood. Therefore, this study was conducted to characterize the impact of HS on the mechanism of TJ regulation in the porcine jejunum using transcriptomic analysis.

## 2. Materials and Methods

This study was conducted on the premises of the National Institute of Animal Science, adhering to the guidelines set forth by the Institutional Animal Care and Use Committee of the National Institute of Animal Science (No. 2022-0554). All the methods were followed sequentially, except the quantitative validation of real-time PCR, which was performed in our previous study [15].

### 2.1. Animals and Treatments

In this study, a total of 24 barrows (crossbred Yorkshire × Landrace × Duroc; with an average initial body weight of 56.71 ± 1.74 kg) were employed. They were randomly divided into two treatment groups, namely, optimal temperature (NT, n = 8) and HS (n = 8). In NT treatment, an environment with an average temperature of 23 °C and relative humidity of 70% was maintained throughout the experimental period. In HS treatment, an average temperature of 32 °C and relative humidity of 85% was maintained during the feeding trial. The experimental period lasted for 56 days (28 days in the grower phase and 28 days in the finisher phase). A basal diet was formulated to meet or exceed the nutrient requirements outlined by the National Research Council [16] for growing–finishing pigs. The experimental diet formulation is presented in Appendix A. Throughout the study, the pigs were provided ad libitum access to water and feed.

### 2.2. Sample Collection 

At the conclusion of the feeding trial, within the experimental period, three pigs from each treatment group were selected and euthanized for gut sample collection. The abdominal cavity was promptly opened, and the intestinal tissue segments were carefully extracted and rinsed with ice-cold phosphate-buffered saline (PBS). Approximately 500 mg of jejunal tissue, from the midpoint of the total length of the small intestine, was collected and immediately flash-frozen in liquid nitrogen. These samples were stored at −80 °C until required for subsequent analysis.

### 2.3. Transcriptomic Analysis

Transcriptomic data were employed and reprocessed as described in our previous study [15]. Total RNA was isolated using TRIzol™ reagent, for transcriptomic analysis (Invitrogen, Carlsbad, CA, USA). RNA quality was evaluated using an Agilent 2100 Bioanalyzer (Agilent Technologies, Amstelveen, The Netherlands), and RNA quantification was performed using an ND-2000 Spectrophotometer (Thermo Inc., Madison, WI, USA). Libraries were prepared from total RNA using the NEBNext Ultra II Directional RNA-Seq Kit (New England Biolabs Ltd., Hitchin, UK). mRNA was isolated according to the manufacturer’s instructions using the Poly(A) RNA Selection Kit (LEXOGEN, Inc., Vienna, Austria) and used for cDNA synthesis and fragmentation. Indexing was performed using Illumina indexes 1–12, and a PCR was used for the enrichment process. The mean fragment size of the libraries was determined using a TapeStation HS D1000 Screen Tape (Agilent Technologies, Amstelveen, The Netherlands). Quantification was performed using a library quantification kit on a StepOne Real-Time PCR System (Life Technologies, Grand Isle, NY, USA). High-throughput sequencing (paired-end 100 sequencing) was performed using NovaSeq 6000 (Illumina Inc., San Diego, CA, USA). FastQC [17] was used for the quality control of raw sequencing data. Adapter sequences and low-quality reads (<Q20) were removed using FASTX_Trimmer [18] and BBMap [19]. The cleaned reads were mapped to the NCBI Genome *Sus scrofa* using TopHat [20]. Read Count (RC) data were normalized using the FPKM + Geometric method in EdgeR (R software V 3.6.0) [21]. Fragments Per Kilobase of transcript per Million mapped reads (FPKMs) values were calculated using Cufflinks [22]. Data analysis and visualization were performed with ExDEGA (Excel-based Differentially Expressed Gene Analysis) software (ebiogen Inc., Seoul, Republic of Korea).

### 2.4. Bioinformatic Analysis 

ExDEGA software, developed by ebiogen (Seoul, Republic of Korea), was used for the initial processing and analysis of differentially expressed genes (DEGs). Subsequently, DEGA was conducted using ExDEGA v1.6.8 software, applying a cut-off at a normalized gene expression (log2) of 4 and a *p*-value < 0.05. DEGs were identified based on a ≥2.0-fold change observed in transcript levels. Following DEG filtering, gene ontology (GO) annotation analysis was conducted using the DAVID bioinformatics program (https://david.ncifcrf.gov, accessed on 3 March 2024) for gene identification and annotation. The RNA-Seq data were analyzed using the Kyoto Encyclopedia of Genes and Genomes (KEGG) database (www.genome.jp, accessed on 3 March 2024).

### 2.5. RNA Extraction and Quantitative Real-Time PCR

Total RNA was extracted from the liquid nitrogen-frozen jejunum samples using TRIzol Reagent (Takara Bio Inc., Shiga, Japan). RNA integrity and quality were evaluated by 1% agarose gel electrophoresis and spectrophotometric analysis at optical density 260/280, respectively. cDNA was synthesized using a commercial reverse transcription (RT) kit (Takara Bio Inc.). The cDNA products were then stored at −20 °C for subsequent relative quantification by PCR. All reverse-transcription quantitative polymerase chain reactions (RT-qPCR) were performed in triplicate, using SYBR Green kits on an Applied Biosystems ABI 7500 system (Thermo Fisher Scientific, Waltham, MA, USA). The expression levels of the genes were normalized by glyceraldehyde 3-phosphate dehydrogenase (GAPDH) using the 2^−ΔΔCT^ method. The primer sequences used for RT-qPCR are listed in Table 1.

### 2.6. Statistical Analysis

Pigs from each treatment group were used as experimental units for gene expression using real-time PCR. Statistical analyses were performed using the GLM procedure in SAS software 9.4 (SAS Inst. Inc., Cary, NC, USA) using Tukey’s multiple comparison test. *p* < 0.05 was considered statistically significant. 

## 3. Results

### 3.1. Analysis of DEGs 

mRNA-sequencing data detected 30,048 genes. A total of 43 DEGs were revealed based on ≥2-fold change, normalized data (log2) ≥ 4, and *p*-value < 0.05 (Appendix A). These DEGs are comprised 28 upregulated and 15 downregulated DEGs and the selected 19 genes relevant to tight junction and actin cytoskeleton formation are presented in Table 2.

### 3.2. Clustering and Functional Enrichment Analysis

Gene expression between the NT and HS treatments was compared. The genes with a ≥2-fold change value, *p* < 0.05, and normalized data (log2) < 4 were used to generate a principal component analysis (PCA), volcano plot, clustering heatmap, and functional enrichment annotation of the treatments. The results of the PCA using DEGs showed that the x-axis and y-axis rates were 91% and 4%, respectively, indicating in each temperature condition a stronger relationship in the x-axis (Principal Component 1, PC1) than in the y-axis (PC2) (Figure 1A), so we decided to compare the groups with respect to HS impact. HS impact highlights two Principal Components explaining 95% of the total variance: 91% of the total variance is explained by PC1, and 4% by PC2. Therefore, the x-axis should be the first principal component indicating a similarity of groups (NT vs. HS) in PC1. The correlation matrix between samples showed that samples within the same group exhibited high correlations. Additionally, the correlations between groups (NT vs. HS) displayed distinct variability, as shown in Appendix A, which supports the observed similarity in PC1 from the PCA analysis. Volcano plot analysis showed that the difference in fold change corresponded to log2 (fold change) (x-axis) and log10 (*p*-value) (y-axis) (Figure 1B). The gene expression patterns of DEGs between the NT and HS treatments are shown in the hierarchical cluster analysis (Figure 1C). An analysis was conducted for each sample in the two treatments, and the results demonstrated that from six samples there were two distinct gene expression patterns associated with the temperature conditions.

Functional enrichment analyses of the jejunum were performed based on the GO classification and KEGG database (Figure 1D). Enrichment was expressed as biological process (BP), cellular component (CC), molecular function (MF), and KEGG pathway, and the top 10 GO classifications were selected. BPs were enriched in nine pathways, viz., bleb assembly, muscle structure development, actin filament organization, negative regulation of complement activation, T cell-mediated immunity, response to heat, sarcomere organization, muscle contraction, and negative regulation of vascular smooth muscle cell proliferation. In CC, DEGs were mainly enriched in seven pathways, viz., stress fiber, Z-disc, cytoplasm, actin cytoskeleton, actin filament, cytolytic granules, and filamentous actin. MFs were enriched in eight pathways, viz., actin binding, structural constituents of muscle, pyridoxal phosphate binding, actin filament binding, calmodulin binding, muscle alpha-actinin binding, glutathione transferase activity, and metal ion binding (including calmodulin binding). KEGG pathway analysis revealed that the DEGs were related to five pathways, viz., vascular smooth muscle contraction, motor proteins, hypertrophic cardiomyopathy, dilated cardiomyopathy, and actin cytoskeleton regulation.

### 3.3. Protein-Protein Interaction (PPI) Network Analysis

The network for the interaction between genes and DEGs of NT and HS treatments is presented in Figure 2. According to gene interaction network analysis, significant gene interactions were associated with myosin light chain 9 (*MYL9*). Moreover, there were eight significant genes interacting with *MYL9.*

### 3.4. Key Candidate Genes Involving in Actin Cytoskeleton and Filament Formation Pathway

We conducted validation by qPCR selecting 12 upregulated DEGs and two downregulated DEGs involving actin cytoskeleton (actin gamma 2, *ACTG2*; desmin, *DES*; glycogen phosphorylase, *PYGM*; *MYL9*; myosin light chain kinase, *MYLK*; protein phosphatase 1 regulatory inhibitor subunit 14A, *PPP1R13A*; tropomyosin 1, *TPM1*; tropomyosin 2, *TPM2*; calponin1 1, *CNN1*; PDZ and LIM domain 3, *PDLIM3*; PDZ and LIM domain 7, *PDLIM7*; Purkinje cell protein 4, *PCP4*; phosphoenolpyruvate carboxykinase 1, *PCK1*; and phosphoenolpyruvate carboxykinase 2, *PCK2*) along with five genes associated with TJs and adherens junctions (*ZO1*, *CLDN1*, *CLDN3*, *CLDN4*, and *OCLN*) in porcine jejunal tissue (Table 2).

### 3.5. Quantitative Real-Time PCR Validation of Gene Expression

The relative mRNA levels of the selected genes were compared at different temperatures (NT vs. HS) (Figure 3). Among the selected genes, 14 of the 19 genes revealed significant differences between the NT and HS groups. Response to HS in genes related to gut integrity resulted in downregulation of *ZO1*, *CLDN1*, and *OCLN* compared to the NT condition. Regarding actin cytoskeleton and filament, HS treatment revealed significantly higher relative mRNA levels of *ACTG2*, *DES*, and *MYL9* than the NT treatment (*p* < 0.01). Furthermore, the HS group was significantly upregulated in *TPM1*, *TPM2*, *CNN1*, *PDLIM3*, and *PCP4* compared to the NT group (*p* < 0.05). However, the relative mRNA levels of *PCK1* and *PCK2* presented significantly downregulated expression in the HS group compared to those in the NT group (*p* < 0.05).

## 4. Discussion

The dominant contributing variables as gene transcriptions were in the pigs exposed to high thermal temperatures. These were indicated by exclusive eigenvalues in the equation of the scores: −1.707 (NT-1), −1.293 (NT-2), −3.304 (NT-3), −0.325 (HS-1), 0.813 (HS-2), 1.815 (HS-3). The PCA analysis also clearly distinguished environmental thermal temperature, indicating significant genetic differences between these two groups. However, significant variation within the group was also observed, confirming previous reports that the heat stress response varies within populations due to underlying genetic variation [23,24,25,26].

High temperature conditions in the housing environment affect the health, production, and welfare of farm animals [26,27,28]. Pigs, in particular, are susceptible to HS owing to their poorly developed sweat glands and thick subcutaneous fat layers, which hinder efficient heat dissipation [29]. HS in pigs causes a range of adverse effects on intestinal morphology, mucosal immunity, integrity, digestive enzyme secretion, and antioxidant status [10]. These adverse effects of HS on animal health and productivity are driven by the disruption of intestinal barrier function, which has been caused under 30–35 °C temperatures [30,31,32]. Hyperthermia compromises the intestinal barrier function, which results in increased permeability [33,34,35]. TJs are primarily composed of structural and functional proteins. Structural proteins including *OCLN*, members of the *CLDNs*, and junctional adhesion molecules (JAMs) form the core structural framework. Intestinal epithelial cells are connected by intercellular TJ proteins that regulate paracellular permeability and are essential for maintaining the integrity of the epithelial barrier [36].

The function of these proteins is altered by HS, which downregulates the expression of TJ proteins. In this study, the relative mRNA levels of TJ proteins (*ZO1*, *CLDN1*, and *OCLN*) were downregulated under HS conditions. *ZO1* is a plaque protein that binds to other proteins, creating a scaffold, or interacts with specific transmembrane proteins to anchor them to the cytoskeleton [37]. *CLDN1* forms the structural foundation of TJs by sealing the gaps between adjacent epithelial cells [38,39]. Furthermore, *OCLN*, containing one intracellular loop and two neutral extracellular loops, plays a critical role in sealing adjacent cells, forming a barrier against macromolecules and small ions [40]. In terms of gene expression, these functional proteins in the intestinal epithelial cells showed differences in response to HS. This seems to be affected by the short- or long-term impact of heat exposure time. The short-term impact of HS in the porcine jejunum (7–21 h exposure) led to reduced *ZO1* expression, but *CLDN1* expression increased [41]. However, some studies reported that porcine jejunum exposed to HS (>33 °C) for over 28 days increased the relative mRNA expression of TJ proteins containing *ZO1*, *CLDN1*, and *OCLN* compared to that at normal temperature (22 °C), which was in accordance with our results [42,43]. 

We identified DEGs involved in actin cytoskeleton and myosin filament organization pathways in intestinal epithelial cells (Figure 4). In this study, porcine jejunal tissues exposed to chronic HS revealed significantly upregulated relative mRNA expression of *ACTG2*, *DES*, *MYL9*, *MYLK*, *TPM1*, *TPM2*, *CNN1*, *PDLIM3*, and *PCP4*, whereas the expression of *PCK1* and *PCK2* was downregulated. These genes are associated with the integrity and permeability of the intestinal barrier. The epithelial barrier function is regulated by apical junctions and specialized structures within the plasma membrane. These structures are composed of adhesive and scaffolding proteins anchored to diverse cytoskeletal structures such as actin filaments, intermediate filaments, and microtubules [44]. Cytoskeletal proteins are essential for the differentiation, directional migration, and continuous renewal of epithelial cells along the crypt–villus axis [45]. Additionally, the actin cytoskeleton plays a critical role in regulating junctional integrity and remodeling under physiological and pathological conditions. Consequently, the expression of cytoskeletal protein genes, including *ACTG2*, *TPM1*, *TPM2*, *MYL9*, and *MYLK*, exposed to high temperatures might be increased, as HS induces the reorganization of apical junctions in jejunal epithelial barrier dysfunction. 

*ACTG2* participates in gastrointestinal peristalsis by regulating cell motility and maintaining the cell skeleton, both of which are related to the contraction ability of epithelial cells [46]. Consistent with our results, a previous study reported that *ACTG2* expression was negatively correlated with the height of the epithelium in the small intestine of rabbits [47]. Furthermore, intestinal damage from ischemia reperfusion injury leads to increased serum concentration of *ACTG2* in rats [48].

The predominance of differentially expressed proteins associated with the cytoskeleton indicated the impact of HS on cellular architecture and motility. It was reported that *MYL9*, involved in cell motility and contraction, is upregulated by HS, resulting in increased intestinal permeability in pigs [49]. *MYLK*, also known as *MLCK,* regulates TJ permeability and catalyzes *MYL9* phosphorylation [49,50,51]. In agreement with this study, previous studies have reported that HS induces intestinal barrier permeability with the upregulated expression of *MYLK* in pigs and rats [49,52]. *TPM1* and *TPM2* are isoforms of tropomyosin that are essential for regulating the assembly, stability, and motility of intestinal epithelial cells [53]. The upregulation of proteins such as *ACTG2*, *MYL9*, *TPM1*, *TPM2*, and *MYLK* in response to HS likely indicates the coordinated modulation of actin cytoskeleton dynamics, adversely affecting intestinal integrity. *DES* is a cytoskeleton protein that participates in the construction of the cytoskeletal structure and muscle contraction as a major intermediate filament, resulting in the integrity and rigidity of muscle cells [54,55]. Previous studies have shown that proteins interact with heat shock proteins (HSPs), that can be induced by HS [56]. Similar to that in our study, another study reported that HS in porcine skeletal muscle significantly increased the mRNA expression of *DES* [57]. The upregulated *PCP4*, which is involved in acquired immunity and the resistance to pathological microbiota infection [58], may indicate an activated immune system with increased intestinal permeability resulting from HS. There are two unique isoforms of phosphoenolpyruvate carboxykinase, viz., the cytoplasmic isoform (*PCK1*) and the mitochondrial isoform (*PCK2*). *PCK1* is a crucial enzyme in gluconeogenesis and catalyzes the conversion of oxaloacetate to phosphoenolpyruvate, a key intermediate in the glycolytic and gluconeogenic pathways [59,60]. Downregulation of *PCK1* may result from disorders of glucose and lipid metabolism [61]. Pigs with intrauterine growth retardation (IUGR) had significantly lower *PCK1* mRNA levels in the jejunum and decreased *CLDN* expression compared to healthy pigs [62]. To normalize mitochondrial function, the biosynthesis of glycerophospholipids, the main components of biomembranes, is indispensable. Mitochondrial morphology and function affect the phospholipid composition of the membrane [60,63]. Thus, *PCK2* plays an important role in the regulation of glycerol phosphate synthesis [64]. The downregulation of *PCK2* expression may be caused by a decrease in glycerophospholipid metabolism. Recent studies have demonstrated that depressed *PCK2* levels in different tissues, such as the porcine jejunum and hepatocytes, result from mycotoxin exposure [65,66]. Furthermore, the relative intensity of intestinal TJ proteins such as *CLDN4*, *ZO1*, and *OCLN* in the deoxynivalenol-supplemented group decreased [66]. This study showed that HS affects porcine jejunal mucosal gluconeogenesis by decreasing mucosal glucose homeostasis and energy metabolism in the cytoplasm and mitochondria. 

## 5. Conclusions

We identified 43 DEGs in the porcine jejunum exposed to HS, and these genes were associated with the regulation of the actin cytoskeleton and filament-binding in the KEGG pathway. Furthermore, genes involved in TJ, actin cytoskeleton, and filament formation revealed changes in their relative mRNA expression. Based on these outcomes, our findings suggest that HS induces depression in gut integrity, which may be related to genes involved in the actin cytoskeleton and filaments of cellular components in growing–finishing pigs.

## Figures and Tables

**Figure 1 animals-14-02476-f001:**
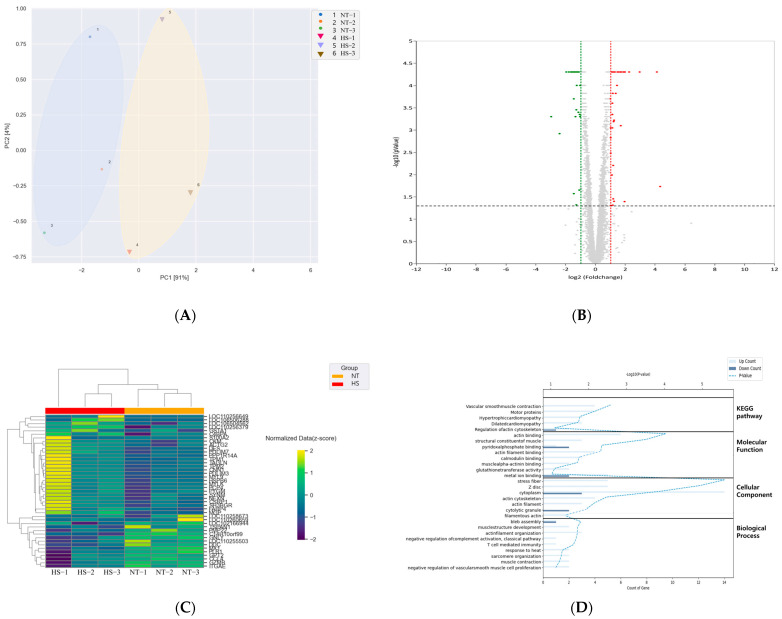
Summary of the DEGs (DEG; *p* < 0.05, ≥2.0-fold change) in the porcine jejunum under HS temperature. (**A**) PCA based on DEGs of the NT and HS groups; (**B**) volcano plot of HS and NT groups; (**C**) clustering heatmap of DEGs in HS and NT groups; (**D**) functional annotation chart of the DEGs in NT and HS groups. NT, normal temperature; HS, heat stress.

**Figure 2 animals-14-02476-f002:**
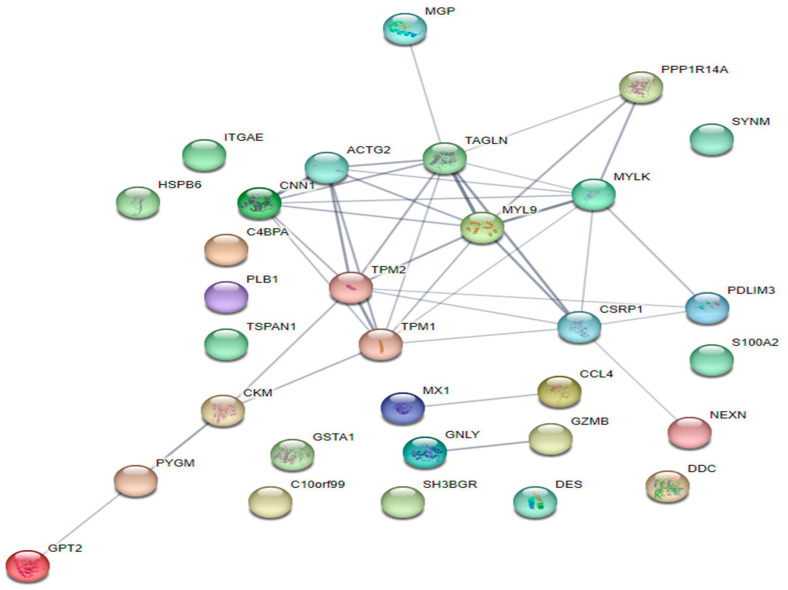
Network analysis of the effect of heat on porcine jejunum in the DEGs of transcriptomic analysis using the Search Tool for the Retrieval of Interacting Genes (STRING). The network displays the interaction between DEGs based on HS and NT (fold change ≥ 2.0; normalized data (log2) ≥ 4; and *p* < 0.05).

**Figure 3 animals-14-02476-f003:**
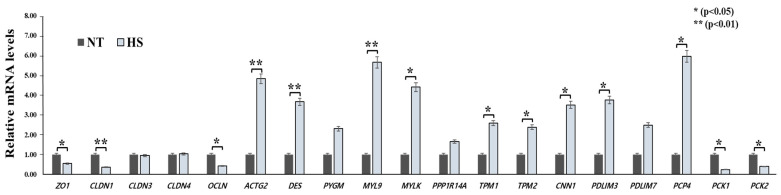
Impact of HS on relative mRNA levels in selected genes. * indicates a significance of *p* < 0.05; ** indicates a significance of *p* < 0.01. NT, normal temperature; HS, heat stress.

**Figure 4 animals-14-02476-f004:**
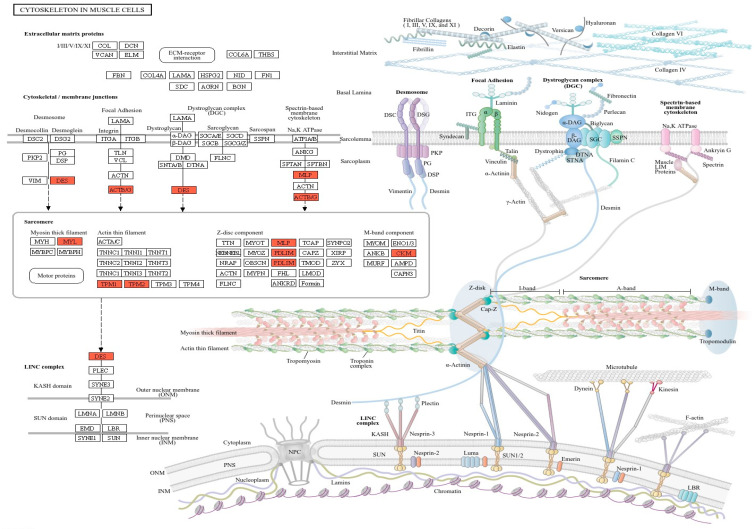
KEGG pathway of cytoskeleton in muscle cells.

**Table 1 animals-14-02476-t001:** Primer sequences of genes selected for analysis by real-time PCR.

Gene	Primer Sequence (5′-3′)	Product Size (bp)	GenBank No.
*ZO1*	F GGGGCAATCTCAACTCCTGT R GGTTGTCCAACTTGGGCAT	137	XM 021098896.1
*CLDN1*	F TTTCCTCAATACAGGAGGGAAGC R CCCTCTCCCCACATTCGAG	74	NM_001244539.1
*CLDN3*	F GCCAAGATCCTCTACTCCGC R GAGAGCTGCCTAGCATCTGG	197	NM_001160075.1
*CLDN4*	F CTCTCGGACACCTTCCCAAG R GCAGTGGGGAAGGTCAAAGG	192	XM_005661969.2
*OCLN*	F TCTCAGCCAGCGTATTCTTTC R GCACATCACGATAACGAGCAT	111	XM 005672525.3
*ACTG2*	F CCTGGCATTGCCGACAGGAT R GGCCAGGATAGAGCCTCCGA	123	XM_021087371.1
*DES*	F TTCCGAGCGGATGTGGATGC R CTGAAGCTGGGCCTGCAGTT	135	NM_001001535.1
*PYGM*	F GTGGAGATGGCGGAAGAGGC R TCGATGACGTGCCGAAGCTC	140	XM_003122588.5
*MYL9*	F CACCAAGAAGCGGCCACAGA R TGCCCTCCAGGTACTCGTCC	188	NM_001244472.1
*MYLK*	F ATCAGGGAGTCCCGCCACTT R CTGCTGTGCAGGTGGCTTCT	142	XM_001929078.6
*PPP1R14A*	F TGAGCAAGCTGCAGTCTCCG R CGGTACAGCTCCTCCAAGCG	163	NM_214337.1
*TPM1*	F TGAGCTGGTGTCGCTGCAAA R ATCGGCTTCAGCATCGGTGG	130	NM_001097483.2
*TPM2*	F GATGTGGCCTCCCTGAACCG R TTCTCGGCCTCCTCCAGCTT	103	NM_001129947.1
*CNN1*	F GCAGGAGCAGGAGCTTCGAG R CTGGTGCCAGTTCTGGGTGG	166	NM_213878.1
*PDLIM3*	F GCTGGCGGCACTCAGAAGAT R TTCGCAGTACAGCTCCCCCT	174	NM_001001637.1
*PDLIM7*	F AGCAGAATGGACAGCCGCTC R GAGGATGCGGAAGGAACGGG	122	XM_021084391.1
*PCP4*	F GCTGGGGCCACCAATGGAAA R TGAGACTGAATGGCCACCGC	128	XM_021071001.1
*PCK1*	F TGCTCCCGGAACCTCTGTGA R GGTCGATGATGGGGCACTGG	107	NM_001123158.1
*PCK2*	F TCTTTCGGCAGCGGCTATGG R ACATAGCGCTTCTTCCCCGC	152	NM_001161753.1
*GAPDH*	F CCACGGTCCATGCCATCACT R GCCTGCTTCACCACCTTCTTG	268	XM_021091114.1

**Table 2 animals-14-02476-t002:** Information table for DEGs selected.

Gene ID	Gene Name	Universal Gene Name
100736682	*ZO1*	zonula occludens-1
100625166	*CLDN1*	claudin 1
431781	*CLDN3*	claudin 3
733578	*CLDN4*	claudin 4
397236	*OCLN*	occludin
100520667	*ACTG2*	actin gamma 2, smooth muscle
396725	*DES*	desmin
733659	*PYGM*	glycogen phosphorylase
100157760	*MYL9*	myosin light chain 9
396848	*MYLK*	myosin light chain kinase
397610	*PPP1R14A*	protein phosphatase 1 regulatory inhibitor subunit 14A
100037999	*TPM1*	tropomyosin 1
396693	*TPM2*	tropomyosin 2
396911	*CNN1*	calponin 1
414421	*PDLIM3*	PDZ and LIM domain 3
100624749	*PDLIM7*	PDZ and LIM domain 7
110256491	*PCP4*	Purkinje cell protein 4
100144531	*PCK1*	phosphoenolpyruvate carboxykinase 1
403165	*PCK2*	phosphoenolpyruvate carboxykinase 2

## Data Availability

All figures and tables used to support the results of this study have been included.

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
