# Peer review of "Heat Stress Induces Alterations in Gene Expression of Actin Cytoskeleton and Filament of Cellular Components Causing Gut Disruption in Growing–Finishing Pigs"

_animals, 2024, doi:10.3390/ani14172476_

Round 1

Reviewer 1 Report

Comments and Suggestions for Authors

The paper presents studies of the effect of heat stress on the mechanism of TJ regulation in the pig jejunum using transcriptomic analysis. Heat stress has an extremely negative impact on pigs, affecting not only productivity traits, but also generally suppresses the immune system and increases the animals' susceptibility to infectious diseases. In this regard, the presented studies are relevant and can serve as the basis for further research into the effect of heat stress on the physiology of animals.

Materials and methods are described in detail, and additional files are provided. The results are quite logical and concise.

There are some questions and comments regarding the work.

1) Line 82-84 “In HS treatment an average temperature of 32 °C and relative humidity of 85% was maintained during the feeding trial.”

Why an average temperature of 32°C was chosen for HS. How can this be substantiated, are there studies confirming that this temperature corresponds to heat stress in pigs.

2) Line 163-166 “The results of PCA using DEGs showed that the first and second contribution rates were 91% and 4%, respectively, indicating in each temperature condition a stronger relationship in the second condition than in the first (Figure 1A) "

What do you mean by “second condition”.

3) Clustering by temperature is not evident, neither in PCA (Fig. 1A) nor in the clustering heatmap of DEGs in HS and NT groups (Fig. 1C). Three clusters are clearly visible in Figure 1C: HS-1; HS-2 and HS-3; NT-1, NT-2 and NT-3. Explanation needed as to why this happened.

4) Line 154-157 “A total of 43 DEGs were revealed based on ≥ 2-fold change, normalized data (log2) ≥ 4, and p-value <0.05. These DEGs are presented in Table 2 and comprised 28 upregulated and 15 downregulated DEGs that are presented in Table S1.”

Please be more specific about what is presented in Table 2.

Author Response

Comment: The paper presents studies of the effect of heat stress on the mechanism of TJ regulation in the pig jejunum using transcriptomic analysis. Heat stress has an extremely negative impact on pigs, affecting not only productivity traits, but also generally suppresses the immune system and increases the animals' susceptibility to infectious diseases. In this regard, the presented studies are relevant and can serve as the basis for further research into the effect of heat stress on the physiology of animals.

Materials and methods are described in detail, and additional files are provided. The results are quite logical and concise.

There are some questions and comments regarding the work.

Response: We are very thankful to the reviewer for their deep and thorough review and especially the point comments that helped us to fully understand all their comments clearly. Certainly, our authors tried to revise the whole manuscript several times to minimize any mistakes according to your comments.

Comment 1: Line 82-84 “In HS treatment an average temperature of 32 °C and relative humidity of 85% was maintained during the feeding trial.”

Why an average temperature of 32°C was chosen for HS. How can this be substantiated, are there studies confirming that this temperature corresponds to heat stress in pigs.

Response 1: Thank you for pointing it with your comment. The impacts of heat stress in pigs happen with various phenotypes such as not only a decreased growth rate and feed intake but also impaired intestinal functions. There are enough studies demonstrating that heat stress under 30-35 °C leads to reduced feed intake and growth rate with bad intestinal functions. Furthermore, our previous studies regarding the impacts of heat stress in pigs showed reduced villus height and nutrient digestibility losing tight junctions and increasing metabolic markers like cortisol when pigs were exposed to high thermal conditions at an average of 32 °C. We have attached relevant references below and put an explanation in the manuscript (Line 234).

GodyÅ„, D.; Herbut, P.; Angrecka, S.; Corrêa Vieira, F.M. Use of different cooling methods in pig facilities to alleviate the effects of heat stress—a review. Animals. 2020, 10, 1459-1482.

Ortega, A.D.S.V.; Szabó, C. Adverse effects of heat stress on the intestinal integrity and function of pigs and the mitigation capacity of dietary antioxidants: a review. Animals. 2021, 11, 1135-1150.

Comment 2: Line 163-166 “The results of PCA using DEGs showed that the first and second contribution rates were 91% and 4%, respectively, indicating in each temperature condition a stronger relationship in the second condition than in the first (Figure 1A) "

What do you mean by “second condition”.

Response 2: Thank you for your comment. We have revised it at Lines 164-66.

Comment 3: Clustering by temperature is not evident, neither in PCA (Fig. 1A) nor in the clustering heatmap of DEGs in HS and NT groups (Fig. 1C). Three clusters are clearly visible in Figure 1C: HS-1; HS-2 and HS-3; NT-1, NT-2 and NT-3. Explanation needed as to why this happened.

Response 3: We appreciate your deep insight and comments about it. Principal Component Analysis (PCA) is a statistical technique used to transform high-dimensional data into lower dimensions in order to identify and visualize the main patterns in the data. PCA provides a method to reduce the dimensionality while preserving as much variance in the data as possible. In our results, the x-axis explains 91% of the total expression data analysis, while the y-axis explains 4%. Therefore, the x-axis represents the first principal component, and we could confirm the similarity of groups (NT vs. HS) in PC1. A detailed explanation is provided (Lines 164-168).

Comment 4: Line 154-157 “A total of 43 DEGs were revealed based on ≥ 2-fold change, normalized data (log2) ≥ 4, and p-value <0.05. These DEGs are presented in Table 2 and comprised 28 upregulated and 15 downregulated DEGs that are presented in Table S1.”

Please be more specific about what is presented in Table 2.

Response 4: Thank you for your valuable comments. The corrections have been made (Line 156-158).

Reviewer 2 Report

Comments and Suggestions for Authors

The manuscript investigated the impact of heat stress (HS) on the expression of tight junction (TJ) proteins and interaction between genes affecting intestinal barrier function in the porcine jejunum. It is well known that acute heat stress impairs the intestinal barrier function, particularly tight junction (TJ) depression. However, the mechanisms by which heat stress affects TJ regulation are not fully understood especially in case of pigs. Therefore, the manuscript addresses a gap in knowledge.

In the current work, twenty-four pigs were placed in different temperatures (normal and heat stress) for 56 days. At the end of the experiment, jejunal samples were collected from three pigs per treatment and transcriptome analysis was done. Overall, the manuscript is important and very well written. I do not have many major concerns with this work. Below are some comments in the spirit of helping the authors improve their manuscript.

Materials and methods

1. Page2, line 79-80: The authors indicated three treatment groups; optimal temperature (NT, n = 8) and HS (n 80 = 8). Kindly mention the third one. Throughout the manuscript, I could see only two groups. Kindly clarify.

2. Kindly mention the composition and nutritional values of the basal diet.

Comments on the Quality of English Language

English language looks fine.

Author Response

Comments: The manuscript investigated the impact of heat stress (HS) on the expression of tight junction (TJ) proteins and interaction between genes affecting intestinal barrier function in the porcine jejunum. It is well known that acute heat stress impairs the intestinal barrier function, particularly tight junction (TJ) depression. However, the mechanisms by which heat stress affects TJ regulation are not fully understood especially in case of pigs. Therefore, the manuscript addresses a gap in knowledge.

In the current work, twenty-four pigs were placed in different temperatures (normal and heat stress) for 56 days. At the end of the experiment, jejunal samples were collected from three pigs per treatment and transcriptome analysis was done. Overall, the manuscript is important and very well written. I do not have many major concerns with this work. Below are some comments in the spirit of helping the authors improve their manuscript.

Response: We are very thankful to the reviewer for their deep and thorough review and especially the point comments that helped us to fully understand all their comments clearly. Certainly, our authors tried to revise the whole manuscript several times to minimize any mistakes according to your comments.

Comment 1: Page2, line 79-80: The authors indicated three treatment groups; optimal temperature (NT, n = 8) and HS (n 80 = 8). Kindly mention the third one. Throughout the manuscript, I could see only two groups. Kindly clarify.

Response 1: Thank you for pointing this. We apologize for it confusion. The treatments of this experiment were comprised two groups: optimal temperature (NT) and high thermal temperature causing heat stress (HS). The corrections have been made (Line 80).

Comment 2: Kindly mention the composition and nutritional values of the basal diet.

Response 2: Thank you for your comment. The basal diet was a commercial diet meeting or exceeding the nutrient requirements by NRC (2012) for growing-finishing pigs which are already mentioned in Lines 85-87. We have attached a supplemental table to present the experimental diet formulation with nutrient composition (Line 87).

Reviewer 3 Report

Comments and Suggestions for Authors

Report on the manuscript animals-3113126 entitled: Heat stress induces alterations in gene expression of actin cytoskeleton and filament of cellular components causing gut disruption in growing-finishing pigs.

The manuscript is well-written, but some major concerns must be discussed:

-          L. 78. Why were only barrows considered? What about gilts?

-          L. 82-83. Could the difference between humidity be another effect?

-          L. 90. Why were only 3 animals considered? Why not the 8 that were previously selected?

Actually, this is the most concerning/critical issue.

In fact, Figure 1A classification lacks scientific validity. The “n” must be increased. There is a huge difference between samples.

In addition, please increase the quality of the image because the symbols in the legend do not match those in the figure.

-          Figure 3. No SD, RMSE or SEM values are shown. How were the degrees of significance obtained?

-          L. 79. “employed”

Author Response

Comments: Report on the manuscript animals-3113126 entitled: Heat stress induces alterations in gene expression of actin cytoskeleton and filament of cellular components causing gut disruption in growing-finishing pigs.

 The manuscript is well-written, but some major concerns must be discussed:

Response: We are very thankful to the reviewer for their deep and thorough review and especially the point comments that helped us to fully understand all their comments clearly. Certainly, our authors tried to revise the whole manuscript several times to minimize any mistakes according to your comments.

Comment 1: L. 78. Why were only barrows considered? What about gilts?

Response 1: Thank you for your deep review and valuable comments. In fact, the trials on growth performance and physiological impacts under heat stress in pigs were conducted in both barrows and gilts. However, due to the limited number of pigs available for sampling and slaughter, we aimed to minimize variation by using only barrows for the transcriptomics trial. We would like to say that sorry for not considering gilts.

Comment 2: L. 82-83. Could the difference between humidity be another effect?

Response 2: Thank you for your review. Temperature and humidity are values that depict the integrated effects of air temperature and humidity associated with the level of heat stress. High temperature and high relative humidity (and the temperature and humidity index) exacerbate heat stress impacts and are commonly used to quantify the degree of heat stress on animals. Therefore, we treated pigs with both higher temperature and humidity than the normal temperature group to induce heat stress.

Comment 3: L. 90. Why were only 3 animals considered? Why not the 8 that were previously selected?

Response 3: We had numbers of limited slaughter due to animal welfare issues, and had replicates as three pigs per treatment. We also considered the biological variation resulting in the differentially expressed genes (DEGs). However, the number of replicates for statistical analysis is still useful in three replicates, which were referred to in previous studies (Lim et al. 2017; Srikanth et al. 2020). Furthermore, this study showed novel and reliable results in the DEGs according to heat stress impacts. Therefore, we strongly believe that our results show an understanding of heat stress responses in porcine gut.

Lim, K.S.; Lee, K.T.; Park, J.E.; Chung, W.H.; Jang, G.W.; Choi, B.H.; Hong, K.C.; Kim, T.H. Identification of differentially expressed genes in longissimus muscle of pigs with high and low intramuscular fat content using RNA sequencing. Anim. Genet. 2017, 48, 166-174.

Srikanth, K.; Park, J.E.; Ji, S.Y.; Kim, K.;H.; Lee, Y.K.; Kumar, H.; Kim, M.; Baek, Y.C.; Kim, H.; Jang, G.W.; Choi, B.H.; Lee, S.D. Genome-wide transcriptome and metabolome analyses provide novel insights and suggest a sex-specific response to heat stress in pigs. Genes 2020, 11, 540.

Comment 4: Actually, this is the most concerning/critical issue.

In fact, Figure 1A classification lacks scientific validity. The “n” must be increased. There is a huge difference between samples.

Response 4: We appreciate your deep insight and comments about it. Principal Component Analysis (PCA) is a statistical technique used to transform high-dimensional data into lower dimensions in order to identify and visualize the main patterns in the data. PCA provides a method to reduce the dimensionality while preserving as much variance in the data as possible. In our results, the x-axis explains 91% of the total expression data analysis, while the y-axis explains 4%. Therefore, the x-axis represents the first principal component, and we could confirm the similarity of groups (NT vs. HS) in PC1.

Comment 5: In addition, please increase the quality of the image because the symbols in the legend do not match those in the figure.

Response 5: Thank you for your review. The images have 3000 pixels which means high definition and over the minimum required for Animals.

Comment 6: Figure 3. No SD, RMSE or SEM values are shown. How were the degrees of significance obtained?

Response 6: Thank you for your comment. The SEM of relative mRNA levels presented in Figure 3 has been expressed.

Comment 7: L. 79. “employed”

Response 7: The correction has been made (Line 79).

Round 2

Reviewer 3 Report

Comments and Suggestions for Authors

Report on the reviewed version of manuscript animals-3113126 entitled: Heat stress induces alterations in gene expression of actin cytoskeleton and filament of cellular components causing gut disruption in growing-finishing pigs.

There are two main concerns that must be discussed before being considered for publication:

-          The fact that the authors only consider three samples remains a major problem. Although the authors say that three samples provide valuable information and that low “n” is used in other publications, it is still of some concern that in Figure 1A, the authors describe PC1 as the classifying PC by showing 91% of the variability. However, samples 1 and 5, while correctly classified by PC1 in NT and HS, show a large difference from the rest because of PC2 (which the authors indicate explains only 4% of the variance).

What is the cause of the difference between these two samples (1 and 5) and the rest?

With that difference, is it possible that the difference between PC1 and PC2 is biased since there are four samples (2, 3, 4, and 6) that are very different from two (1 and 5)?

-          Figure 3.

The authors have included the SEM of their analyses. Nevertheless, what is going on with PYGM, PPP1R14A and PDLIM7? Why no statistically significant difference was detected? One can understand CLDN3, CLDN4 but…

Please, review, explain and improve the Results description and Discussion if needed.

Author Response

Comment 1: There are two main concerns that must be discussed before being considered for publication:

 - The fact that the authors only consider three samples remains a major problem. Although the authors say that three samples provide valuable information and that low “n” is used in other publications, it is still of some concern that in Figure 1A, the authors describe PC1 as the classifying PC by showing 91% of the variability. However, samples 1 and 5, while correctly classified by PC1 in NT and HS, show a large difference from the rest because of PC2 (which the authors indicate explains only 4% of the variance).

What is the cause of the difference between these two samples (1 and 5) and the rest?

With that difference, is it possible that the difference between PC1 and PC2 is biased since there are four samples (2, 3, 4, and 6) that are very different from two (1 and 5)?

Response 1: Thank you for your comments and we agree with your concern. We tried to have made efforts to address it. We hope our explanations clarify the issue.

PCA is a method of dimensionality reduction that facilitates easier interpretation of data, and when PC1 accounts for over 90% of the variance in the data, it suggests that most of the variability can be explained in one dimension. While the analysis results are not considered problematic, visually, it appears that there is variability within the same group along PC2 (even though it is only 4%), which raises concerns.

We analyzed the correlation matrix data between samples and groups (Figure S6), supporting the PCA analysis results by showing high correlations within the same group and distinct differences with comparatively low correlations between groups (Lines 168-171). In our view on the cause of the difference between two samples (1 and 5), they have relatively high correlations compared to another samples in the group but it still has clear different color between groups (NT vs. HS). Furthermore, rather than focusing solely on the number of dots, it's important to observe whether there are many dots that are dispersed outward. A higher number of such dispersed dots can affect the correlation. We anticipate that the number of dispersed dots may be more on these samples than others. We hope our explanations are enough to address your concern and we strongly believe that the results of our study will provide valuable and useful information as a reference to other researchers future, even if some results might not be entirely clear.

Comment 2:

- Figure 3.

The authors have included the SEM of their analyses. Nevertheless, what is going on with PYGM, PPP1R14A and PDLIM7? Why no statistically significant difference was detected? One can understand CLDN3, CLDN4 but…

Response 2: Thank you for your comments. While PYGM, PP1R14A and PDLIM7 may appear to have high SEM, which suggests a significant difference in appearance, it is possible that no statistically significant difference is detected due to considerable variation within specific samples. Upon re-evaluating the data, we found that the SEM of these genes is actually slightly lower, and the variation may have led to the absence of significant difference.

Comment 3: Please, review, explain and improve the Results description and Discussion if needed.

Response 3: The additional explains have been added to improve the description of the results (Line 168-171).

Round 3

Reviewer 3 Report

Comments and Suggestions for Authors

Authors can continue to go back and forth with superficial responses to reviewers' comments but not consider making any modifications to their work.

It remains a concern to me that the study was conducted with only 3 samples.

I would recommend that the authors include the equation from their PCA analysis and discuss it to clarify the very peculiar classification of their samples.

Author Response

Comment 1: Authors can continue to go back and forth with superficial responses to reviewers' comments but not consider making any modifications to their work.

It remains a concern to me that the study was conducted with only 3 samples.

I would recommend that the authors include the equation from their PCA analysis and discuss it to clarify the very peculiar classification of their samples.

Response 1: Thank you for your comment and we understand your concern. We made some sentences in the discussion part to deal with the issue (Line 166-174; 234-240). Please note that the primary findings of this manuscript are not limited to the results of the PCA analysis; rather, the PCA results are presented simply to observe variability using dimensionality reduction